# Sleep Problems and Disabilities of the Arm, Shoulder, and Hand in Persons with Thoracic Outlet Syndrome—A Cross-Sectional Study

**DOI:** 10.3390/ijerph191912486

**Published:** 2022-09-30

**Authors:** Natasa Milenovic, Aleksandar Klasnja, Renata Skrbic, Svetlana Popovic Petrovic, Sonja Lukac, Gordana Devecerski

**Affiliations:** 1Faculty of Medicine, University of Novi Sad, 21 137 Novi Sad, Serbia; 2The Special Hospital for Rheumatic Diseases, 21 137 Novi Sad, Serbia; 3Oncology Institute of Vojvodina, 21 204 Sremska Kamenica, Serbia; 4Center of Radiology, University Clinical Center of Vojvodina, 21 137 Novi Sad, Serbia; 5Department at the Clinic for Medical Rehabilitation, University Clinical Center of Vojvodina, 21 137 Novi Sad, Serbia

**Keywords:** sleep quality, thoracic outlet superior, pain

## Abstract

Thoracic outlet syndrome (TOS) arises as a result of a specific relationship among the anatomical structures that may cause compression in the muscles, nerves, and/or blood vessels in the neck, thereby compromising the local circulation. The aim of the current study was to establish the presence of sleep disturbance and disability in the shoulder, arm, and hand in individuals affected by TOS, as well as to ascertain if there are any differences in these findings relative to TOS-free individuals. The study sample comprised 82 TOS patients and 81 TOS-free individuals aged 19–66 years. Data were gathered by administering the Disabilities of the Arm, Shoulder, and Hand (DASH) and Pittsburgh Sleep Quality Index (PSQI) instruments. The results showed that both the DASH (*t* = −13.21, *p* < 0.001) and PSQI (*t* = −7.27, *p* < 0.001) scores obtained by the TOS group were higher relative to the controls and were strongly and positively correlated (*ρ* = 0.58, *p* < 0.01). As positive DASH scores may be indicative of TOS, they signal the need for further diagnostic evaluations. In individuals in whom TOS is already diagnosed, high DASH scores imply that further sleep quality assessments are required, as compromised sleep patterns may undermine quality of life.

## 1. Introduction

Thoracic outlet syndrome (TOS) results from the concomitant occurrence of several etiological factors that are a consequence of disturbance of the anatomical–topographic relations in the thoracic outlet, including the clavicular space, anterior scapular opening, and mechanics of rib joints and cartilage, leading to the narrowing of the space through which neurovascular structures pass, resulting in their compression and thus irritation of neurovascular structures [1]. TOS is estimated to affect 1 in 100,000 individuals of the general population, but it is extremely rare in children [2], while being more common among women compared to men (with reported ratios of 4:1 and 2:1) [3]. The symptoms of TOS are complex and may manifest as pain in the neck region, numbness along the arms, hand numbness, impaired gross motor strength of the arm muscles, loss of sensation in the hands, cold arms and hands, poor circulation in the hands, pain in the hands, chest pain, throbbing sensations, palpitations, shoulder pain, headaches, dizziness, tinnitus, and loss of consciousness [4,5]. As their presence compromises the affected individual’s quality of life (QoL) [6,7], particularly sleep quality [8,9], further assessments are required, given that sleep is a physiologically unconscious state, and its main function is to revitalize the body and conserve energy [10].

Modern trends in physical medicine consider not only the cause-and-effect relationship of certain conditions and diseases, as well as ways of treating such conditions, but also focus on any resulting reduction in QoL. In the available literature, sleep and sleep quality are mainly observed through the prism of other diseases and conditions [11,12]. To date, no research has been conducted linking TOS with sleep problems. Unlike TOS, other similar conditions—such as neck and lumbar syndrome and fibromyalgia, which are accompanied by pain and disability—have been investigated in the context of sleep disorders [9,13,14]. However, several authors have recognized the possibility of sleep disturbances being caused by TOS. In order to assess the QoL and functional level of TOS patients, Vastamaki et al. proposed the use of a questionnaire specifically designed for this population, including an item probing into sleep problems [15]. However, the number of studies focusing on the symptoms, etiology and diagnosis of TOS has grown rapidly in recent years [1,3,15,16,17,18].

On the contrary, the Disability of Arm, Shoulder and Hand (DASH) questionnaire has been widely used to assess the functional incapacity of the arm, shoulder and hand in people with TOS [18,19,20], and the available findings indicate that reduced hand functionality significantly affects work activity and daily functioning, as well as lowers the global QoL of these individuals [7]. The dynamic lifestyle, as well as the complexities of functioning in the modern social and work environment, necessitate the adoption of a more comprehensive view of the disease, which must be examined from the physical, mental, and social aspects. Guided by this premise, the aim of the present study was to assess sleep quality and arm, shoulder, and hand functions in individuals diagnosed with TOS. Its further objective was to identify any differences in these findings compared to individuals unaffected by TOS, as well as to determine whether there is a connection between sleep problems and disability.

## 2. Materials and Methods

The study sample comprised 163 individuals, 81 of whom were TOS-free and 82 had a TOS diagnosis. All of the study participants resided in the municipality of Novi Sad, Vojvodina Province, Republic of Serbia. The required sample size (*n* > 60) was calculated on the basis of the known prevalence of TOS in the general population (4%) and the number of inhabitants (600,000) in the South Bačka district, which includes the city of Novi Sad and its surroundings, with 95% confidence level and a 5% margin of error. All of the participants voluntarily signed the informed consent form. The study was approved by the Ethics Committee of the Faculty of Medicine, University of Novi Sad.

As indicated in Table 1, while women (*n* = 123) predominated in the overall sample relative to men (*n* = 40), the gender distribution in the TOS and TOS-free groups was comparable (*χ*^2^ = 1.29; *p* = 0.279).

Other descriptive characteristics of the sample are given in Table 2. Respondents with TOS were, on average, older than those in the TOS-free group, had a higher body weight, and had a higher body mass index.

This cross-sectional study was conducted between 2014 and 2015 and involved individuals in whom TOS was diagnosed in accordance with pertinent literature [11,21]. All of the TOS patients were treated in the outpatient clinic of the Special Hospital for Rheumatic Diseases, Novi Sad. The TOS patients were sent by their family doctor for a specialist examination by a physiatrist due to existing complaints related to the neck, shoulder, arm, or hand. During the study period, they were not engaged in any form of physical therapy. 

The inclusion criteria for the TOS group were as follows:Confirmation of a TOS diagnosis based on the presence of at least three of the following symptoms: Pain in the neck region, numbness along the arms, hand numbness, impaired gross motor strength of the arm muscles, loss of sensation in the hands, cold arms and hands, poor circulation in the hands, pain in the hands, chest pain, throbbing sensation, palpitations, shoulder pain, headache, dizziness, tinnitus, and loss of consciousness.Positive findings on at least one of the provocative tests (Adson, Wright, Roos, Halstead, Elvi, and costoclavicular/military brace test) [4,22].Positive hand oscillography findings [23].Evidence of hyperplasia of the transverse extension of the seventh cervical vertebra or cervical rib on a standard radiological image of the cervical spine [24].

The following exclusion criteria were applied when forming the study sample: Carpal tunnel syndrome, lateral and/or medial epicondylitis, complex regional pain syndrome, Horner’s syndrome, Raynaud’s syndrome, cervical disc protrusion, brachial plexus trauma, any systemic–immune disease, deep vein thrombosis in the upper extremities, shoulder joint instability, lung and kidney diseases, hyperthyroidism (which can affect sleep), any acute infectious disease, malignant diseases, previous surgical treatment aimed at alleviating TOS, and use of benzodiazepines and antidepressants in the preceding month.

The TOS-free control group comprised adult volunteers recruited for the study by their family doctor. For inclusion in the TOS-free group, absence of any TOS symptoms and negative provocative test results were mandatory, along with the absence of any exclusion criteria adopted for the TOS group.

The relevant data were gathered by verifying the TOS diagnostic criteria and by administering the relevant questionnaires to both TOS patients and TOS-free individuals that formed the control group. A detailed description of all adopted diagnostic procedures is provided in Appendix A.

Instruments. The Disabilities of the Arm, Shoulder and Hand (DASH) questionnaire. This instrument, originally developed by Hudak and colleagues [25], was adopted for determining the degree of dysfunction in the upper extremities based on symptom evaluation and tridimensional scoring—comprising the physical, social, and psychological domains—reflecting the respondent’s ability to perform activities of everyday life. When completing the DASH questionnaire, the relevant aspects of daily life and symptomology are assessed using a five-point Likert scale. The Serbian version of this instrument (both long and short) was developed by Dr Tomislav Palibrk and his colleagues from the Orthopedic and Traumatology University Clinic, Clinical Center of Serbia [26]. When the instrument was applied to our sample, high reliability was attained (α = 0.98).

Pittsburgh Sleep Quality Index (PSQI). This instrument was developed by Buysse and colleagues in 1988 at the Faculty of Medicine in Pittsburgh, US [27]. It was adopted in this study, as its authors’ aim was to (a) provide a reliable, valid, and standardized evaluation tool; (b) differentiate individuals that sleep well from those that do not; (c) offer an index that can be easily applied in both clinical practice and research; (f) allow clinicians to evaluate different sleep disturbances that can impact on the sleep quality. The Serbian version of the PSQI demonstrated good reliability and validity [28]. The questionnaire comprises 19 self-rated items, along with five that are answered by the respondent’s bed partner or roommate (if available). The ratings obtained on these final five items are not considered when calculating the PSQI score, which comprises the following seven components: Sleep duration, sleep disturbances, sleep latency, daytime dysfunction due to inadequate sleep, habitual sleep efficiency, subjective sleep quality, use of sleeping medication.

Each of these seven components can generate a score of 0–3 depending on the response provided, resulting in a final PSQI score of 0–21, whereby higher scores reflect lower sleep quality and those above 5 indicate sleep disturbance. The Cronbach’s Alpha in our study was adequate (α = 0.85).

All data obtained in the present study were analyzed using the SPSS 19.0 (SPSS Inc., Chicago, IL, USA) statistical package. Prior to the analyses, the raw data were coded and input into a database specifically designed for this purpose. Descriptive statistics included means and standard deviations, which were calculated for numerical data, while frequencies and percentages were reported for attributive features. The parametric Student’s *t*-test and univariate analysis of covariance (ANCOVA) were performed to determine the significance of differences between the observed features. The correlations between the individual variables were assessed via Spearman’s and partial correlations. Differences between groups, and in relation to categorical variables, were evaluated via Pearson’s chi-squared test. The reliability of the scales was checked by the value of the Cronbach’s alpha coefficient. The statistical significance level was set as *p* < 0.05.

## 3. Results

### 3.1. Descriptive Statistics and Differences between Groups

The descriptive statistics related to the DASH and PSQI scores obtained by the TOS and the control TOS-free group are reported in Table 3.

According to the findings yielded by the Student’s *t*-test, the mean DASH scores were significantly higher in the TOS group relative to the TOS-free group (*t* = −13.21, *p* < 0.001). A one-way ANCOVA, in which age was adopted as a covariate, revealed that the difference between the compared groups was still statistically significant (*F*(1, 160) = 31,021.12, *p* < 0.001, partial eta squared = 0.46). The TOS group also scored higher on all 30 individual DASH items (Appendix B, Table A1).

According to the mean PSQI scores, the individuals assigned to the TOS group had a significantly lower sleep quality relative to the TOS-free group (*t* = −7.27, *p* < 0.001). The one-way ANCOVA results obtained after controlling for the effect of age showed that the difference between the compared groups remained statistically significant (*F*(1, 160) = 344.95, *p* < 0.001, partial eta squared = 0.20).

As shown in Table 4, all seven PSQI components yielded significantly higher scores for the TOS group, reflecting poorer sleep quality. Consequently, the overall PSQI score was also higher in the TOS group.

A mean sleep duration of 6.16 h (SD = 1.24) was obtained for the TOS group, compared to 6.55 (SD = 1.02) for the TOS-free group, and this difference was statistically significant (*t* = 2.18; *p* = 0.03). As noted earlier, PSQI scores above 5 indicate sleep disturbance and, based on this criterion, 57 (69.5%) and 22 (28.2%) of the respondents in the TOS and TOS-free groups, respectively, were affected by sleep disturbance. Once again, this difference was statistically significant (*χ*^2^ = 47.59; *p* < 0.001).

### 3.2. Correlation between Sleep Problems and Disabilities of the Arm, Shoulder, and Hand

Using Spearman’s correlation coefficient, the correlation between the participants’ BMI and age and their DASH and PSQI scores was analyzed, and the findings are reported in Table 5.

As can be seen from Table 5, a statistically significant positive correlation was obtained in the TOS group between age and DASH score, as well as between the scores obtained on the DASH and PSQI questionnaires. In other words, those respondents with TOS who had a higher level of arm, shoulder, and hand disability also experienced greater sleep problems. The partial correlation established after controlling for the significance of age indicates the presence of a statistically significant association between DASH and PSQI scores (*r*_partial_ = 0.39, *p* < 0.001). In the TOS-free group, no correlation was found between age and DASH or PSQI scores, but the scores obtained on these instruments were positively correlated. 

## 4. Discussion

Thoracic outlet syndrome (TOS) arises as a result of a specific relationship among the anatomical structures that may cause compression in the muscles and other anatomical structures (nerves and/or blood vessels) in the neck, thereby compromising the local circulation. While TOS can manifest as a range of symptoms, our investigation indicated the presence of a link between sleep disturbance and disabilities of the arm, shoulder, and hand, as issues with sleep quality were much more prevalent in the TOS group compared to the TOS-free group. Although sleep disorders and compromised upper extremity function are not exclusively related to TOS, but also affect individuals with other cervical problems [29,30], these issues cause significant problems for patients affected by TOS, as they adversely affect their daily functioning.

However, the link between sleep disturbance and TOS has never been investigated, even though symptoms associated with sleep problems due to pain include daytime fatigue and drowsiness, poor sleep quality, sleep latency, and decreased cognitive and motor functioning [31,32]. It should be noted that pain in the arm can contribute to sleep irregularities [33,34], while some authors also attribute sleep disorders, depression, and anxiety to neck pain [35], which was also reported by our respondents.

In adults, sleep insufficiency is defined as sleep of duration below the seven to eight hours per night deemed optimal for revitalizing the body and conserving energy [36]. In our study, the average sleep duration of 6.16 h obtained for the TOS group based on the PSQI questionnaire responses was significantly shorter than the 6.55 h noted for the TOS-free group. This difference can be attributed to the effect that TOS has on sleep quality, which was confirmed by all domains of this instrument, resulting in shorter and less restful sleep compared to the TOS-free group. Extant research also shows that a less than optimal sleep duration is associated with musculoskeletal pain [37]. It is believed that due to the activation of the sympathetic nervous system and the inhibition of muscle relaxation, muscle tone increases, and so does the risk of pain [38]. The relationship between sleep and pain is bidirectional, given that sleep disorders can be caused by pain, whereas interrupted or insufficient sleep can lower the pain threshold and exacerbate spontaneous pain [39]. Therefore, it is necessary to simultaneously treat both pain and sleep disorders in people with TOS and other conditions accompanied by chronic pain. Chronic neck pain is associated with poor sleep quality, as well as tiredness upon waking, and leads to a decreased daily efficacy that may cause psychological problems [40]. People with chronic pain have difficulty falling asleep and maintaining sleep.

In our study, respondents with TOS reported greater sleep latency, which can be attributed to arm or shoulder pain, but also difficulties in positioning the head, neck, and spine when falling asleep. Certain positions, in combination with various spinal diseases, can also cause sleep problems [41]. Available evidence also suggests that the positioning of the head, neck, and body when sleeping in a certain position can be directly responsible for compressing neurovascular structures, leading to increased sleep latency or difficulties upon waking up, including headaches, fatigue, and inefficacy [42]. Sleep problems and pain in the neck and shoulder region can also be linked to inadequate pillow support during sleep [43].

Whether they had trouble falling asleep, were waking up during the night or sleeping less, or had some other manifestation of sleep disorders, the TOS subjects had a greater need for taking sleeping medication. In the pertinent literature, various drug treatments for sleep problems associated with chronic pain are discussed, including nonsteroidal anti-inflammatory drugs, benzodiazepines, and opioid analgesics [44]. However, there is a prevalent view among researchers that sleep problems in people with chronic pain are not approached systematically, in terms of evaluation, treatment and follow-up, and there is no clearly defined approach or treatment strategy.

All of the aforementioned symptoms of poor sleep quality can be related to the combined compression of neurovascular structures in the upper thoracic outlet region, primarily due to reduced circulation in the shoulder girdle, neck, and head region [43,45], which can be induced by provocative hand positions during sleep, but can also be caused by the very nature of TOS. Thus, sleep problems can be explained by the pathophysiology of anatomical structures, which also leads to other symptoms, such as tinnitus, dizziness, fatigue, fainting, loss of vision, and headaches [45,46,47]. 

Our analyses further revealed that individuals suffering from TOS have a significantly higher degree of shoulder, arm, and hand disability, as reflected by their scores on all 30 DASH items, as well as their overall DASH score. Interestingly, differences in the responses provided by the TOS and TOS-free groups were noted not only in relation to the questions pertaining to elevating the arm above the shoulder, neck, and head level, but also to questions related to other everyday activities, such as writing, unlocking the door, preparing meals, opening heavy doors, gardening or agricultural work, making the bed, carrying a shopping bag or a handbag, and using a kitchen knife. TOS was also found to affect their ability to partake in certain hobbies and sports. We expected that individuals with TOS would find it challenging to perform activities that required elevating their arms to or above the neck and head level, such as changing lightbulbs, washing their back, washing and drying hair, or putting on a sweater, but they also reported difficulties in relation to all DASH items. In accordance with their physical status, i.e., the anatomical–topographical relationship of the region’s structures, those respondents affected by TOS reported severe weakness of the arms, shoulders and hands, as well as stiffness in these regions, along with the associated pain when performing various daily activities. All of these issues made them feel considerably less capable, confident, or useful. 

According to the findings yielded by other studies in which DASH was administered to individuals diagnosed with TOS, their scores ranged from 38 to 55 [19,48,49]. The average DASH score for our TOS group was 39, which is in accordance with the values reported by other authors. However, as none of these patients were candidates for surgery, as expected, higher DASH scores were noted in studies involving surgical patients. Thus, given that DASH is one of the most commonly used instruments for monitoring functional changes in both surgically and conservatively treated patients [48,49,50], we believe that its application is of great importance in identifying and diagnosing TOS. 

In a study conducted by Ohman et al., DASH was found to be useful in identifying the link between BMI and reduced hand, shoulder, and body functioning in people with TOS [48]. These authors noted that overweight individuals (BMI = 25.1–30.0) had the lowest DASH scores compared to respondents in all other BMI categories (underweight, normal, and obese groups—BMI > 30.1). In our sample, no correlation between BMI and the total DASH score was obtained. It is also noteworthy that the BMI was significantly lower in the TOS-free group, whereby the BMI obtained for the TOS group was in the overweight range.

In this study, a high positive correlation was obtained between the PSQI and DASH scores in those subjects with TOS syndrome, while the correlation was of medium strength in the TOS-free individuals. According to these findings, in the prognostic sense, people who have a higher degree of arm, shoulder, and hand disability may be predisposed to greater sleep problems. Moreover, individuals with a positive DASH score should be diagnostically evaluated in terms of establishing a TOS diagnosis. On the contrary, individuals that have been diagnosed with TOS and have high DASH scores should have their sleep quality assessed.

The therapeutic approach in the treatment of TOS should be multidisciplinary and comprehensive [16]. Conservative treatment strategies should ideally combine physical (diathermy, laser, TENS, and ultrasound) and medicinal approaches (starting with non-steroidal anti-inflammatory drugs and/or opioids, and progressing to miorelaxants, anticonvulsants, anticoagulants, pregbalin, and eventually botulinum toxin injections), with emphasis on daily application of kinesitherapy (postural and ergonomic education) [51,52,53].

When interpreting the reported findings, some limitations in the present study should be noted. Specifically, given that only age and gender were considered in the analyses, other relevant sociodemographic information should be examined in future investigations, including educational attainment, occupation and workplace environment, and physical activity level (sports, hobbies, etc.). A further limitation arises from the somewhat older TOS cohort relative to the participants in related studies, which reduced the utility of comparisons with the findings reported in the extant literature. However, the age group considered in the present study was governed by the demographic composition of patients referred for treatment at the institution where the research was conducted. Moreover, as women predominated in the TOS group, there is a possibility that menopause-related issues affected their sleep, which was another factor omitted from the analyses. Finally, although the clinical diagnostic procedures adopted in this study conform with the prevalent guidelines, and standardized, highly reliable self-assessment instruments were used to gather pertinent data, in future investigations, it would be beneficial to supplement this methodology with modern imaging methods in order to obtain more objective findings. 

## 5. Conclusions

This research marks the first attempt to link TOS with sleep problems and upper limb disability. The results obtained in this research confirmed the initial assumption that people with TOS have significantly more pronounced sleep problems compared to those unaffected by TOS, suggesting that they need to be closely monitored. Moreover, further research into sleep quality in people with TOS is needed to assess our findings and provide further insights into this scantly explored phenomenon.

## Figures and Tables

**Table 1 ijerph-19-12486-t001:** Sample characteristics by group (TOS/TOS-free) and gender.

Gender	TOS-Free	TOS	Total
Male, *n*, (%)	23 (28.4%)	17 (20.7%)	40
Female, *n*, (%)	58 (71.6%)	65 (79.3%)	123
Total, *N*, (%)	81 (100%)	82 (100%)	163

TOS = thoracic outlet syndrome.

**Table 2 ijerph-19-12486-t002:** Descriptive characteristics of the sample.

	Group	Min	Max	Mean	SD	*t*	*p*
Age (years)	TOS-free	20.00	65.00	44.70	10.70	−4.48	0.000
	TOS	19.00	66.00	51.74	9.35		
Body height (cm)	TOS-free	156.00	195.00	172.22	9.02	2.29	0.023
	TOS	146.00	194.00	168.83	9.89		
Body weight (kg)	TOS-free	50.00	117.00	73.28	17.412	−1.02	0.306
	TOS	47.00	125.00	76.02	16.67		
BMI	TOS-free	17.76	36.33	24.50	4.42	−2.79	0.006
	TOS	17.51	47.05	26.65	5.35		

TOS = thoracic outlet syndrome, BMI = body mass index.

**Table 3 ijerph-19-12486-t003:** Descriptive statistics related to the DASH and PSQI scores.

Group	Instrument	Min	Max	Mean	SD	Skewness	Kurtosis
TOS-free	DASH	0.00	41.67	7.83	9.10	1.58	2.26
	PSQI	1.00	11.00	4.59	2.24	0.80	0.15
TOS	DASH	0.00	81.67	39.59	19.76	0.19	−0.75
	PSQI	2.00	16.00	7.98	3.55	0.36	−0.92

TOS = thoracic outlet syndrome, DASH = Disability of Arm, Shoulder, and Hand Questionnaire, PSQI = Pittsburgh Sleep Quality Index.

**Table 4 ijerph-19-12486-t004:** Differences in the scores obtained for different PSQI components between TOS and TOS-free group.

Components of PSQI	Group	*N*	Mean	SD	*t*	*p*
C1 Subjective sleep quality	TOS-free	81	0.22	0.69	−3.59	<0.001
	TOS	82	0.74	1.12		
C2 Sleep latency	TOS-free	81	0.86	0.85	−6.52	<0.001
	TOS	82	1.77	0.92		
C3 Sleep duration	TOS-free	81	1.01	0.66	−2.32	0.021
	TOS	82	1.28	0.81		
C4 Habitual sleep efficiency	TOS-free	81	0.22	0.57	−3.22	0.002
	TOS	82	0.59	0.84		
C5 Sleep disturbances	TOS-free	81	1.09	0.45	−7.79	<0.001
	TOS	82	1.83	0.73		
C6 Use of sleep medication	TOS-free	81	0.17	0.52	−2.04	0.044
	TOS	82	0.39	0.81		
C7 Daytime dysfunction	TOS-free	81	1.01	0.37	−4.93	<0.001
	TOS	82	1.38	0.56		
Global PSQI Score	TOS-free	81	4.59	2.24	−7.27	<0.001
	TOS	82	7.98	3.56		

PSQI = Pittsburgh Sleep Quality Index, TOS = thoracic outlet syndrome.

**Table 5 ijerph-19-12486-t005:** DASH and PSQI scores in relation to age and BMI.

	Group	Age	BMI	DASH
BMI	TOS-free	−0.01		
	TOS	0.11		
DASH	TOS-free	0.15	0.12	
	TOS	0.31 **	−0.01	
PSQI	TOS-free	0.07	0.03	0.40 **
	TOS	0.21	0.04	0.58 **

** *p* < 0.01; BMI = body mass index, TOS = thoracic outlet syndrome, DASH = Disability of Arm, Shoulder, and Hand Questionnaire, PSQI = Pittsburgh Sleep Quality Index.

## Data Availability

Not applicable.

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
