# Peer review of "Sleep Problems and Disabilities of the Arm, Shoulder, and Hand in Persons with Thoracic Outlet Syndrome—A Cross-Sectional Study"

_ijerph, 2022, doi:10.3390/ijerph191912486_

Round 1
Reviewer 1 Report
- Title: I suggest you inform in the title the type of study (a cross-sectional study).
- line 58-60: Include a reference that brings this information.
- line 68-69: Clarify how the sample calculation was made for this study.
- line 294: The ref used is 2003. A more current reference will help in understanding the relevance of this cut proposed by the authors.
- line 345: From what is set out in this paragraph, I suggest briefly including some considerations about therapeutic strategies to address patients. What therapies could be suggested, so that it would help in the management of TOS and concomitantly, in the aid of a more adequate and healthy sleep.
- line 356-360: Sincerely dismay the authors about the limitations presented. The instruments used for evaluation are originally a subjective analysis. This is their characteristic. Perhaps the limitation is that you have not used other forms of quantitative (objective) analysis.
And it has been said that this study has an innovative feature. This is good, but it serves as an incentive for studies to futudeveloped from the methodology used in this study. And no, as a limitation because i did not have another previous study to compare.
Reviewer 2 Report
Milenovic et al. report on patients with thoracic outlet syndrome (TOS), record their disability in the shoulder, arm, and hand (DASH), and habitual sleep quality (PSQI). The cross-sectional study involved patients treated in an outpatient clinic for rheumatic diseases (n=82) and TOS-free individuals that formed a control group (n=81). The results showed that DASH and PSQI scores were higher in the TOS group than in the control group and that DASH scores were positively correlated with PSQI. The authors conclude that further investigation of sleep quality is needed in TOS patients, as impaired sleep patterns may affect quality of life.
The research question is interesting, but it only provides a small contribution to this field. I also have some concerns about the preparation and description of this study and therefore recommend a comprehensive revision and text shortening for publication:
It is not surprising that patients with pain and anatomical functional limitations subjectively report poorer sleep. Furthermore, there is the question of recruitment and more detailed characteristics of the control group, which is not sufficiently described in the manuscript.
Specifically, I have the following comments:
Line 42: “… As their presence compromises the affected individual’s quality of life (QoL)“ à citation missing
Line 43: “To date, no research has been conducted linking TOS with sleep problems, although several authors have recognized the possibility of sleep disturbance being caused by TOS” à really (cf. Vastamaki et al)? ... but at least in patients with similar/comparable symptoms this should be indisputable in terms of content (in the context of pain or in clinical practice).
Line 68: Study sample is not sufficiently described – esp. the TOS-free subsample!
Line 70 (374): Vote from 2014!? – why it takes so long, pls explain …
Line 78: Why are the controls not better aligned (at least for evaluation)?
Line 90 to 160: This section belongs shortened or in the appendix, because the diagnostics of TOS is not content of the manuscript ...
Line 170 and 186 … the same unbelievable high Alpha – the result is difficult to imagine …
Line 193: in Tab5 it is Spearman …
Line 196: accepted (correct wording?)
Tab 3 Formatting (.,), 3 decimal places?, values (PSQI) are not in accordance with Tab. 4
Line 210: Supp Tab A or Tab.1; DASH differentiation / values for domains?
Fig 1: x-Axis – write group name/label instead of 1,00 and 2,00
Tab 4 … p < or exact?
Tab 5 Why the correlation with age and BMI - for control purposes? If the age is significant, you should control or stratify
Other questions for/about the discussion:
What is the control group/comparison?
Is the result TOS-specific or does it only show the association with pain and physical complaints / life circumstances?
Which indicator do the authors equate with quality of life and why - the DASH or the PSQI?
Line 256: „On the other hand, our cohort was older than the participants in other studies on this subject, which can be attributed to the older age of individuals that are treated at the clinic where this research was conducted“ à ... why are these older?
Line 273: “… they also reported difficulties in relation to all DASH items, confirming the high selectivity of this instrument for this group of respondents.” à I don’t understand this statement
Line 277: “ ... along with the associated pain when performing various daily activities.” – results?
Line 281: “Therefore, this instrument can be used in clinical practice as one of the indicators for diagnosing TOS.“ really? - cf. your description in the methods section … the results presented do not necessarily fit with it …
Line 295: results/numbers missing
Line 305: citation missing … and the following literature seems not up to date
Line 333: results/numbers missing
Line 346/347: “both vascular and neurogenic symptoms … our findings indicate that sleep problems in this cohort are primarily linked to vascular TOS” à results/numbers missing
Reviewer 3 Report
This study firstly examined subjectively evaluated sleep in the patients of Thoracic Outlet Syndrome (TOS). The authors administered PSQI and DASH questionnaires to the TOS patients and TOS-free controls to evaluate sleep and disabilities in upper extremities, respectively. Poorer sleep in the TOS patients compared to the controls was suggested by global PSQI score, and significant correlations between DASH score and PSQI score were detected both in the TOS patients and TOSS-free controls. The authors insisted further research to explore sleep quality in the TOS patients.
Although the present study is important as the first study to report sleep in the TOS patients, there are following points to be revised or clarified.
1) I confirmed the Serbian version of the DASH questionnaire in the reference #21. However, there is no description of Serbian translation for PSQI. Has PSQI been validly translated in Servian language?
2) The year(s) to administer the questionnaires must be described.
3) The PSQI scores between Table 3 (Mean: 4.592 and 9.52 in TOS-free and TOS, respectively) and Table 4 (Mean: 4.593 and 7.976 in TOS-free and TOS, respectively) are different. Which is true???
4) In results section, 3.1. and 3.2. should be combined into one section, and I think Figure 1 is not necessary. As it is minor point, the section number describing Table 5 is wrong (now, it is 3.1.).
5) Discussion section must be reorganized to demonstrate the significance of this study, which is firstly examine sleep characteristics in the TOS patients.
(1) From this viewpoint, the 2nd to 5th paragraph, which explain the characteristics of participants looks too long.
(2) In the 6th paragraph, the authors introduced a paper by Ohman et al. that “overweight individuals had the lowest DASH scores compared to respondents in all other BMI categories”. Although the descriptions in this paragraph seem to suggest higher BMI is related to lower DASH scores, the results of Ohman’s paper showed higher DASH scores in obese participants (BMI: >30.1) whose BMI was higher than overweight participants (BMI: 25.1-30).
(3) In the 7th paragraph, it is described that the present TOS patients suffer from sleep disorders, such as sleep apnea, snoring and restless leg syndrome. Although this description seemed to be based on the results of PSQI reported by participants’ bed partner or a roommate, the results are not shown in the results section.
(4) The 9th and 10th paragraph should be combined into one paragraph.
(5) In line 321 in the 11th paragraph, “sleep latency” must be “increased sleep latency”.
(6) In the 12th paragraph, the finding of ref. #37 is important, suggesting insufficient sleep in the TOS patients may induce increased vulnerability to pain, resulting in a vicious circle.
(7) 72 and 79% of the participants in the TOS and TOS-free group were females. Considerable number of the female participants might suffer from menopausal disorders, adversely affecting their sleep. However, there is no description to confirm menopausal disorders or menopausal-related sleep problems for the female participants. If the authors did not confirm menopausal disorders, at least, it should be discussed in discussion section.
Round 2
Reviewer 2 Report
There are still some limitations in the study results and in the study design, which are also justified by the authors. The manuscript has gained significantly in quality through revision. The study is now described in a comprehensible way and the data and results support the authors' conclusion.
Reviewer 3 Report
Revised manuscript is much improved. However, I have found following 2 points to be revised.
1) In Table 2, significant differences between the group were shown in age, body height, and BMI. However, the descriptions in lines# 89-91 explains that body mass and BMI were higher in TOS group.
2) In a sentence in lines# 247-249, “TOS and TOS-free group” must be “TOS group”.
In addition, I recommend the authors to carefully and thoroughly check the manuscript again, and that the manuscript be checked by professional editing service, because I have found following issues, and there may be another issues to be edited or revised.
1) Descriptions in lines# 98-100, where the authors revised, must be reedited.
2) I have never seen the description, “statistically significantly”, which is successive use of 2 adverb in any other scientific papers.
3) In line# 260, “statically” is typo of “statistically”.
4) The authors should consider paragraph break in discussion section, because some paragraphs are very short and should be combined.
